

# Seasonal variations and co-occurrence networks of bacterial communities in the water and sediment of artificial habitat in Laoshan Bay, China

Guangjie Fang[1], Haolin Yu[1], Huaxiang Sheng[1], Chuanxi Chen[2], Yanli Tang[1] and Zhenlin Liang[3]

[1] Fisheries College, Ocean University of China, Qingdao, Shandong, China
[2] College of ocean and earth sciences, Xiamen University, Xiamen, Fujian, China
[3] Marine College, Shandong University, Weihai, Shandong, China

## ABSTRACT

Marine bacteria in the seawater and seafloor are essential parts of Earth's biodiversity, as they are critical participants of the global energy flow and the material cycles. However, their spatial-temporal variations and potential interactions among varied biotopes in artificial habitat are poorly understood. In this study, we profiled the variations of bacterial communities among seasons and areas in the water and sediment of artificial reefs using 16S rRNA gene sequencing, and analyzed the potential interaction patterns among microorganisms. Distinct bacterial community structures in the two biotopes were exhibited. The Shannon diversity and the richness of phyla in the sediment were higher, while the differences among the four seasons were more evident in the water samples. The seasonal variations of bacterial communities in the water were more distinct, while significant variations among four areas were only observed in the sediment. Correlation analysis revealed that nitrite and mud content were the most important factors influencing the abundant OTUs in the water and sediment, respectively. Potential interactions and keystone species were identified based on the three co-occurrence networks. Results showed that the correlations among bacterial communities in the sediment were lower than in the water. Besides, the abundance of the top five abundant species and five keystone species had different changing patterns among four seasons and four areas. These results enriched our understanding of the microbial structures, dynamics, and interactions of microbial communities in artificial habitats, which could provide new insights into planning, constructing and managing these special habitats in the future.

## INTRODUCTION

Global marine biodiversity is threatened by habitat degradation, environmental pollution, overexploitation, and other anthropogenic disturbances (*Timothy, Anne & John, 2006*; *Pereira et al., 2010*; *Todd et al., 2019*). The decline of global biodiversity has resulted in

Corresponding author
Yanli Tang, tangyanli@ouc.edu.cn

changes in marine ecosystem structures (*Verity et al., 2002*). In the coastal waters, habitat loss and degradation are especially tricky problems that must be faced (*Jackson et al., 2001*; *Koldewey et al., 2010*). To recover marine habitats that provide feeding, spawning grounds, and shelters, artificial habitats are constructed to support the functions of coastal ecosystems (*Janiak & Branson, 2021*). Artificial reefs (ARs) offer a series of ecological services like improving marine habitats, increasing fishery resources, and manipulating assemblages of ocean organisms (*Seaman & Sprague, 1993*; *Lima, Zalmon & Love, 2019*), which is an important form of artificial habitat in marine fisheries. ARs facilitate biological production and enhance fishery resources by creating additional habitats for marine organisms (*Ng, Toh & Chou, 2017*). At present, ARs have been deployed worldwide by creating different structures related to fishing, scuba diving and coastal recreation (*Lima, Zalmon & Love, 2019*). In China, the practice of deploying ARs to create artificial habitats has been proposed by the government since the 1970s (*Sun et al., 2017*). The annual investment in ARs is about 100 million dollars in recent years, and its primary purposes are conserving marine environment, enhancing fishery resources, and developing recreational fishing (*Yang et al., 2019*; *Xu et al., 2021*). With the increasing attention to ARs, most studies have focused on the behaviors of targeted species (*Williams-Grove & Szedlmayer, 2017*), designs and engineering (*Woo et al., 2014*), ecological impacts (*Shin et al., 2014*), fishery management (*Lima et al., 2020*) and others (*Chen et al., 2013*). However, few studies focused on the bacterial community structure in ARs.

Microorganisms in marine ecosystems play crucial roles in global biogeochemical processes, such as energy flow, carbon and nutrient cycles (*Steele et al., 2011*; *Sunagawa et al., 2015*). Therefore, recognizing that the functions of microbial communities are essential to master the restoration mechanism of artificial habitats. Scientific attempts have been made to isolate and characterize particular microbiomes to explain the ecological roles of artificial habitats in the 1990s (*Zentgraf et al., 1992*). Then, the dynamics of bacterial communities associated with coral reefs (natural habitats) and artificial habitats were compared to verify the ecological effects of two habitats using terminal restriction fragment length polymorphisms (*Soka, Hutagalung & Yogiara, 2011*). Nowadays, high-throughput sequencing has dramatically facilitated the understanding of the mechanisms of marine microbial ecology (*Langille et al., 2013*). Despite great advancements, studies paying attention to the bacterial communities in the water and sediment of ARs are limited. *Wang et al. (2019c)* assessed the impacts of ARs on bacterial communities in the sediment to reveal the changes of microbial structure and functions. *Qin et al. (2019)* and *Zhu et al. (2020)* studied the shifts of community dynamics and interaction patterns of the protists in Daya Bay after the deployment of ARs.

Microbial community dynamics have been observed at different time scales, from days (*Mangot et al., 2013*), weeks (*Berdjeb et al., 2018*), months (*Marquardt et al., 2016*) to seasons (*Genitsaris et al., 2015*), and years (*Boras et al., 2010*). Apart from seasonality, habitats are important factors that influence microbial communities (*Sun et al., 2019*), especially between water and sediment (*Abia et al., 2017*; *Liu et al., 2018*). After profiling the structures and dynamics of microbial communities, co-occurrence network is a powerful tool to uncover the potential ecological interactions among microorganisms

(*Barberan et al., 2012*), and have been applied to obtain a more integrated understanding of microbial communities (*Mikhailov et al., 2019*; *Zhang et al., 2020a*).

Laoshan Bay is a representative semi-enclosed bay in northern China, with an area of approximately 188 km². Laoshan Bay is an important marine culture area for sea cucumber (*Apostichopus japonicus*) and oyster (*Crassostrea gigas*), and is a stock enhancement area for shrimp and fish species (*Sheng, Tang & Wang, 2018*). The annual mean seawater temperature is approximately 15 °C and has significant seasonal changes (5 °C–25 °C; *Wang et al., 2019b*). Fishery resources in Laoshan Bay have decreased rapidly because of habitat degradation and overexploitation. To help reproductive success and support recruitment, two types of ARs were deployed since 2005: (1) rock reefs were deployed with mean volumes of at least 0.04 m³ (weight 100 kg); (2) concrete reefs were arranged with principal dimensions of 2 m × 2 m × 2 m.

For decades, ARs have been deployed worldwide around coasts to recover marine habitats, and the ecological effects of ARs have been evaluated through periphytons, plankton, benthos, and nekton (*Aleksandrov, Minicheva & Strikalenko, 2002*; *Scott et al., 2015*; *Ng, Toh & Chou, 2017*; *Chen, Yuan & Chen, 2019*). However, few studies have examined the microbial communities. This study elucidated the bacterial communities in the water and sediment of artificial habitat with two types of ARs using 16S rRNA gene sequencing. The major objectives were: (1) to provide a comprehensive understanding of the bacterial community structures and dynamics in the water and sediment of ARs, (2) to reveal the influence of environmental factors on bacterial communities in the water and sediment, respectively, and (3) to analyze the potential interactions among bacteria and identify the keystone taxa in the ARs.

# MATERIALS AND METHODS

## Study sites and sample collection

Triplicate water and sediment samples were collected from Laoshan Bay ARs in January (winter), May (spring), August (summer), and November (autumn), 2020 (Fig. S1). Ninety-six samples from four sampling areas were studied: rock reefs (RR), transition areas (TA), concrete reefs (CR) and adjacent areas (AA). For the water samples, we used a plexiglass to collect 2 L bottom water at every sampling area. For the sediment samples, grab sediment sampler was applied to get about 1 kg surface sediment (0–10 cm) for analysis. Water and sediment samples were stored at a cooler filled with ice and immediately transported back to the laboratory in 2 h. The methods for treating samples were followed as previously described in *Fang et al. (2021)*. All field experiment was permitted by "Measures for annual evaluation and reexamination of national marine ranching demonstration areas", which was promulgated by Ministry of agriculture and rural areas of China.

## Measurements of environmental factors

The temperature (Temp), dissolved oxygen (DO), chlorophyll-a (Chla), sampling depth (Dep), pH and salinity (Sal) of water samples were measured *in situ* using a YSI PRODSS multi-parameter water quality analyzer (YSI, Yellow Springs, OH, USA). Transparency

(Trans) was obtained by a Secchi disk. The turbidity (Turb) of each area was determined by a Turb 430 IR (Xylem Analytics, Weilheim in Oberbayern, Germany). The determination of the total organic carbon (TOC) was performed through a TOC-L series total organic carbon analyzer (Shimadzu, Kyoto, Japan). Eight water environmental factor, including suspended particulate materials (SPM), particulate organic matter (POM), chemical oxygen demand (COD), ammonium ($NH_4$-N), nitrate ($NO_3$-N), nitrite ($NO_2$-N), active silicate ($SiO_3$) and active phosphate ($PO_4$) were analyzed under the guidance of GB/T 12763-2007 (*State Bureau of Quality and Technical Supervision of China, 2007*).

For the measurement of sediment characteristics, Mastersizer 3,000 (Malvern, England, UK) was applied to measure the mean particle sizes (Par). Sediment bulk density (BD) was defined as the dry weight of the sediment (drying at 105 °C for 72 h) divided by the wet volumes. Water content (WC) was the weight proportion of water in the sediment (drying at 70 °C for 72 h). After burning the sediment to ash totally (550 °C for 4 h), organic matter content (OM) was obtained. The method for mud content (MC, dry sediment) was following *Eleftheriou (2013)*. As for the salinity (Sal), electrical conductivity (EC) and pH of the sediment, a mixture of sediment and deionized water (2:5, w/v) was used.

## DNA extraction, PCR amplification, Illumina MiSeq sequencing and Sequence analysis

Genomic DNA of the whole samples was extracted using the FastDNA® SPIN Kit for Soil (MP Biomedicals, Irvine, CA, USA) according to the manufacturer's instructions. Specifically, the universal primers 338F (5′-ACTCCTACGGGAGGCAGCAG-3′) and 806R (5′-GGACTACHVGGGTWTCTAAT-3′) were used, which amplified 468 bp in V3–V4 hypervariable region of 16S rRNA gene. The procedures of the PCR amplification was based on the manufacturer's instructions and the standard protocols of Majorbio Bio-Pharm Technology Co. Ltd. (Shanghai, China). Raw sequence reads were analyzed using QIIME 1.9 (*Caporaso, Kuczynski & Stombaugh, 2010*). The detailed processes were all described in File S1. The sequencing data have been deposited in the National Center for Biotechnology Information (NCBI) Sequence Read Archive database under the accession number PRJNA725051.

## Statistical analysis

The number of OTUs and alpha diversity estimators (Shannon, Simpson, Ace, Chao 1 and Good's coverage) were analyzed among four seasons and two habitats. The seasonal and spatial variations of bacterial community compositions were compared by principal coordinate analysis (PCoA) based on the Bray-Curtis distance matrix (*Borcard, Gillet & Legendre, 2011*). The effects of the temporal and spatial factors on the bacterial communities were tested using permutation multivariate analysis of variance (PERMANOVA) and analysis of similarity (ANOSIM) (*Anderson, 2001*; *Clarke, 1993*). Statistical differences were analyzed using the Kruskal-Wallis test, and the differences were considered significant at $P < 0.05$. The correlations between the most abundant OTUs (relative abundance > 0.5%) and environmental factors, sampling areas were determined
by Spearman correlation analysis, and the visualization was achieved by heatmaps (*Borcard, Gillet & Legendre, 2011*). The impacts of environmental factors on bacterial communities were evaluated by the Mantel test and partial Mantel test (*Smouse, Long & Sokal, 1986*). Statistical analyses were conducted in R 4.0.2 using the "*phyloseq*" and "*vegan*" packages (www.r-project.org).

The visualization of the three co-occurrence networks was built using Cytoscape 3.8.2 (www.cytoscape.org) for the all bacterial communities in the water, sediment and both two habitats. To improve the visibility and sensitivity of the networks, OTUs observed in more than 50% of all samples and the mean relative abundance higher than 0.5% were selected. Then, we calculated the Spearman correlations and significances between OTUs in R. Significant edges ($P < 0.001$) with high correlations ($|r| > 0.8$) were chose to construct the co-occurrence networks. The network topology parameters were calculated using the Analyze Network plugin of Cytoscape to identify the keystone OTUs in the networks (*Cheung et al., 2018*). Three topology parameters were: (1) degree: the number of edges that a node has; (2) betweenness centrality: the number of shortest paths between any two nodes in the graph passing through a node; (3) closeness centrality: the average distance of a node to any other node. After that, min-max scaling was applied to standardize degree, 1-betweenness centrality, and closeness centrality, respectively. Last, keystoneness was defined as the average of degree, 1-betweenness centrality, and closeness centrality after min-max scaling was calculated (*Berry & Widder, 2014*); and the top 10 OTUs with highest scores were identified as the keystone species (*Cheung et al., 2018*). Jaccard index was applied to analyze the similarities of OTUs among networks (*Dmitry et al., 2016*).

# RESULTS

## Bacterial alpha diversities and community compositions

A total of 5,427,513 high-quality bacterial 16S rRNA gene sequences and 17,207 OTUs with 97% similarity levels were identified from 96 water and sediment samples. Sediment samples (15,010 OTUs) had higher OTUs than water samples (9,423 OTUs), and 7,226 OTUs were shared between the two habitats. The Good's coverage was over 90% in all samples, indicating the current sequencing depth was sufficient for this study. Alpha diversity indices showed significant variations between the habitats and seasons (Table 1), while there were no differences among four areas. The diversity indices (Shannon, Chao 1 and Ace) of water samples were significantly lower than sediment, while the diversity indices (Simpson and Good's coverage) of water were higher (Table 1). A seasonal pattern of alpha diversity indices (*e.g.*, Shannon) could be observed in the water samples, which varied from 4.09 to 5.16 (Fig. 1). However, no significant seasonal differences were found in the sediment except for the Simpson diversity index (Table 1).

Seasonal variations of the abundant phyla in the water were more evident than sediment, but the differences among the four areas were low. The total abundance of the three most abundant phyla was over 74% and around 50% for water and sediment samples, respectively. For the water samples, the three most abundant phyla (> 10%) were Proteobacteria, Cyanobacteria, and Actinobacteria, with mean relative abundances of 42.8%, 18.3% and 13.1%, respectively (Fig. 2). Marked seasonal changes were observed in

**Table 1 Number of OTUs and alpha diversity indices in the water and sediment of artificial reefs for four seasons.**

| Season | OTUs | | Shannon | | Simpson | | Ace | | Chao 1 | | Coverage | |
|---|---|---|---|---|---|---|---|---|---|---|---|---|
| | Water | Sediment | Water | Sediment | Water | Sediment | Water | Sediment | Water | Sediment | Water | Sediment |
| Spring | 1726 | 2823 | 5.16 | 6.35 | 0.030 | 0.008 | 4742 | 6875 | 3349 | 5111 | 0.95 | 0.93 |
| Summer | 1227 | 3056 | 4.47 | 6.60 | 0.040 | 0.005 | 3148 | 7210 | 2309 | 5408 | 0.97 | 0.92 |
| Autumn | 984 | 3030 | 4.09 | 6.49 | 0.075 | 0.006 | 2665 | 7373 | 1936 | 5439 | 0.97 | 0.92 |
| Winter | 1273 | 3103 | 4.68 | 6.55 | 0.045 | 0.006 | 3765 | 7490 | 2609 | 5584 | 0.97 | 0.92 |
| Seasonality *P* | 0.001 | 0.492 | 0.001 | 0.322 | 0.019 | 0.030 | 0.001 | 0.055 | 0.001 | 0.124 | 0.001 | 0.136 |
| Water-sediment *P* | 0.004 | | 0.005 | | 0.025 | | 0.007 | | 0.005 | | 0.011 | |

**Note:**
Seasonality *P*: statistical difference among four seasons; Water-Sediment *P*: statistical difference between water and sediment samples. Significant difference was at $\alpha = 0.05$ level.

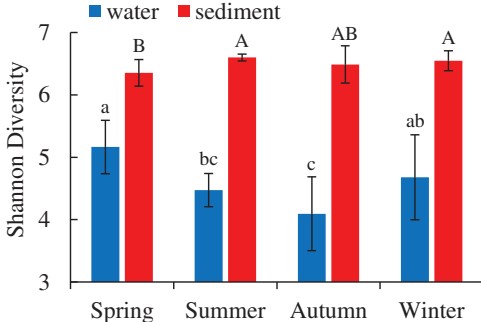

**Figure 1 Seasonal change of Shannon diversity (mean ± SD) of bacterial communities in the two habitats of artificial reefs.** Means with different letter (lowercase letters for water samples, capital letters for sediment samples) are significantly different with *P* value < 0.05.

many bacterial phyla. The phylum Proteobacteria had a relative abundance greater than 56.8% in autumn, while its abundance was 23.6% in summer; the phyla Actinobacteria and Cyanobacteria had the highest abundances in summer (20.9% and 29.4%) and the lowest abundances in winter (6.6% and 13.2%). In the sediment, the bacterial community was dominated (> 10%) by the phyla Proteobacteria, Desulfobacterota, and Acidobacteria, which comprised approximately 28.4%, 10.8%, and 10.2% of the relative abundances, respectively (Fig. 2). The relative abundance of the phylum Chloroflexi (mean abundance was 8.6%, ranked 4th) had marked seasonal changes, whose abundance in autumn (12.0%) was twice that in winter (6.7%).

## Bacterial community structure

Two different bacterial groups were observed between the water and sediment samples (Fig. 3; Table 2). Significant seasonal differences were both observed for the water and sediment samples, and the bacterial communities in the water samples were more divergent and separated than sediment (Fig. S2). The bacterial communities in the sediment varied among seasons, areas, and their interactions based on PERMANOVA and ANOSIM (Table 2). However, there was no significant difference among the four areas in

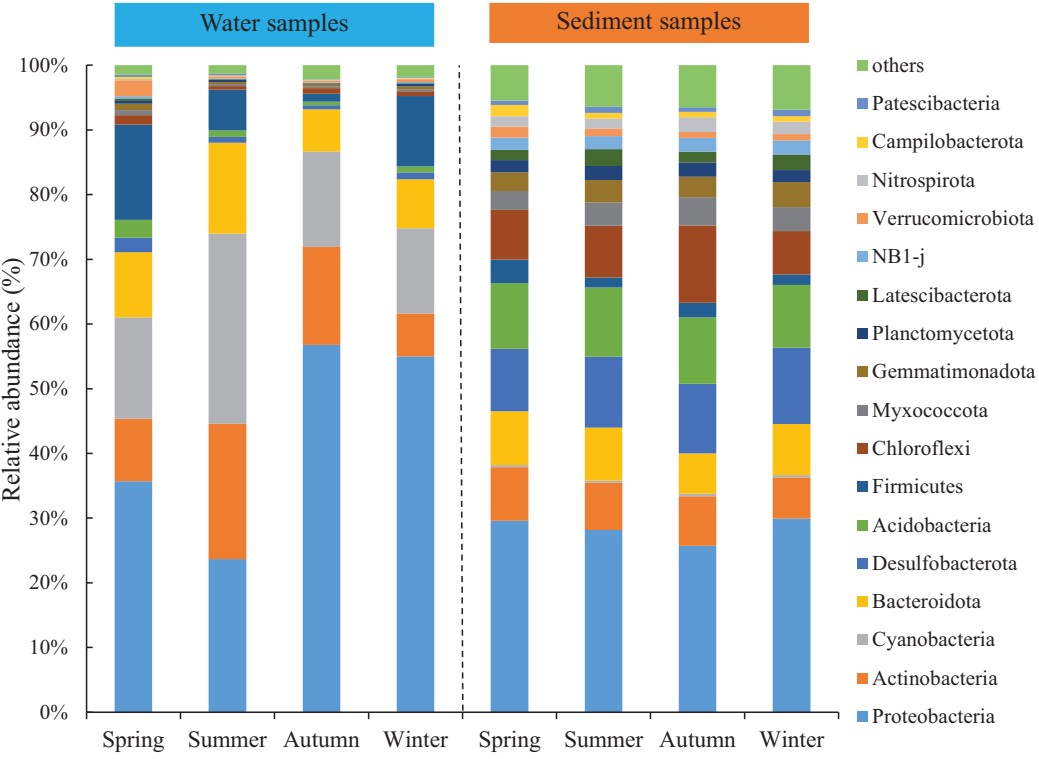

**Figure 2 Relative abundance of bacterial communities at phylum level in the two habitats and four seasons of artificial reefs.**

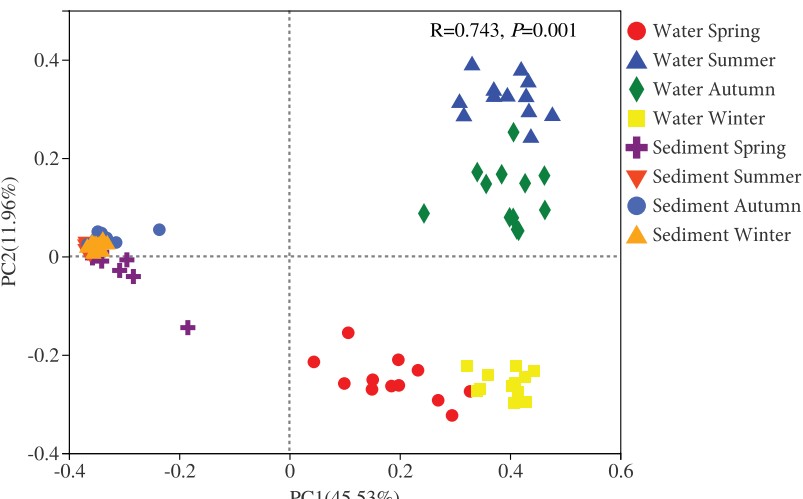

**Figure 3 Principal coordinates analysis (PCoA) plot of bacterial communities in the water and sediment of artificial reefs.** R: the test statistical significance of analysis of similarities (ANOSIM); P: statistical significance value at α = 0.05 level.

the water. The Bray-Curtis dissimilarities among four areas had similar seasonal changing trends in the two habitats; and the dissimilarities were highest in autumn and lowest in summer (Fig. 4). The top 39 abundant water OTUs and 24 sediment OTUs (> 5%), which affiliated to six and thirteen bacterial classes, showed varying abundances among four areas

**Table 2 The effects of habitat, season and site on bacterial communities (OTU level) based on PERMANOVA and ANOSIM. Interaction effects were only calculated by PERMANOVA.**

| Habitat | Effect | PERMANOVA | | ANOSIM | |
|---|---|---|---|---|---|
| | | *F* | *P* | *R* | *P* |
| Total | Habitat | 73.951 | 0.001 | 0.89 | 0.001 |
| | Season | 4.880 | 0.001 | 0.12 | 0.001 |
| | Area | 0.463 | 0.973 | 0.02 | 0.862 |
| | Habitat × Season | 28.950 | 0.001 | – | – |
| | Habitat × Area | 11.096 | 0.001 | – | – |
| | Season × Area | 1.218 | 0.139 | – | – |
| | Habitat × Season × Area | 8.146 | 0.001 | – | – |
| Water | Season | 17.364 | 0.001 | 0.820 | 0.001 |
| | Area | 0.438 | 0.999 | 0.060 | 0.994 |
| | Season × Area | 4.172 | 0.001 | – | – |
| Sediment | Season | 2.905 | 0.001 | 0.227 | 0.001 |
| | Area | 2.163 | 0.001 | 0.133 | 0.001 |
| | Season × Area | 2.689 | 0.001 | – | – |

**Note:**
*F*: the test statistical significance of Permutational multivariate analysis of variance (PERMANOVA); *R*: the test statistical significance of analysis of similarities (ANOSIM); *P*: statistical significance value at α = 0.05 level.

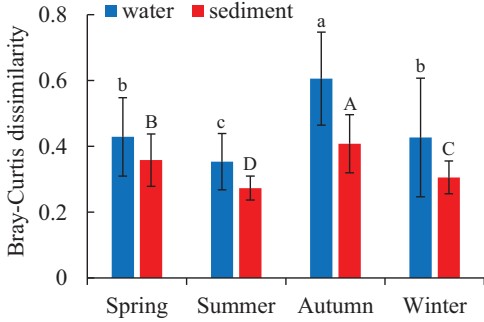

**Figure 4 Seasonal variations of bacterial communities among samples in the water and sediment using Bray-Curtis dissimilarity.** Means with different letter (lowercase letters for water samples, capital letters for sediment samples) are significantly different with *P* value < 0.05.

(Fig. S3). According to the results of cluster trees, bacterial communities in the areas RR and TA were different from areas CR and AA in the water samples (Fig. S3A); communities in the area CR was separate from other three areas in the sediment (Fig. S3B).

## Relationships between abundant OTUs and environmental factors

The relationships among bacterial communities and environmental factors were significant both in the water and sediment samples based on the Mantel test (Table 3, Table S1). For the water samples, $NO_2$-N (r = 0.414) was the most correlating factor in shaping the community; Temp, Trans, $PO_4$, DO, $SiO_3$, and COD were significant correlating factors. In the sediment, MC (r = 0.125) was the only factor that significantly

**Table 3 Relationships between bacterial communities and environmental factors based on the Mantel test and partial Mantel test.**

| Water | | | Sediment | | |
|---|---|---|---|---|---|
| Factor | r | P | Factor | r | P |
| Total | 0.294 | 0.001 | Total | 0.145 | 0.018 |
| $NO_2$-N | 0.414 | 0.001 | MC | 0.125 | 0.050 |
| Temp | 0.346 | 0.001 | | | |
| Trans | 0.287 | 0.001 | | | |
| $PO_4$ | 0.215 | 0.003 | | | |
| DO | 0.181 | 0.004 | | | |
| $SiO_3$ | 0.115 | 0.023 | | | |
| COD | 0.141 | 0.039 | | | |

Note:
r: Statistical significance of Mantel test and partial Mantel test; P: statistical significance value at $\alpha = 0.05$ level. Environmental factors: nitrite ($NO_2$-N); temperature (Temp); transparency (Trans); active phosphate ($PO_4$); dissolved oxygen (DO); active silicate ($SiO_3$); chemical oxygen demand (COD); mud content (MC).

impacted the bacterial community. The relative abundances of the top abundant OTUs were noticeably related to the environmental factors (Fig. 5). For the water samples, thirty-three OTUs were significantly correlated with Temp, while only two OTUs were related to Chla (Fig. 5A). For the sediment samples, thirteen OTUs were highly related to EC, and four OTUs were related to WC and OM (Fig. 5B). Also, two bacterial groups were observed that responded to the environmental factors, conversely. For example, class Gammaproteobacteria and class Alphaproteobacteria were divided into two groups for the sediment samples, correspondingly.

## Co-occurrence networks and keystone species

One co-occurrence network containing total bacteria both in the water and sediment, and two networks for the communities in the water and sediment, were constructed to analyze the connections, stability, and complexity of bacterial communities in the ARs (Fig. 6; Figs. S4 and S5). For the total bacteria network, 185 nodes belonging to 18 phyla and 6,832 edges were detected. A complete distinction of nodes from water and sediment samples formed two modules (Fig. 6A). The most abundant OTUs (top five OTUs with highest abundance) in the network were OTU18562 (Firmicutes), OTU18460 (Cyanobacteria), OTU10090 (Proteobacteria), OTU17592 (Proteobacteria), and OTU18751 (Actinobacteria). Four OTUs mainly from water samples exhibited seasonal variations in their abundances; OTU10090 was from sediment samples, which had no seasonal change (Fig. 7A). Differences of five abundant OTUs among four areas were not significant except OTU18562 (Fig. 8A). The bacterial network for the water samples comprised 189 nodes belonging to 15 phyla, while only 1,772 edges were observed (Fig. S4). OTU18562 (Firmicutes), OTU14763 (Cyanobacteria), OTU17592 (Proteobacteria), OTU18751 (Actinobacteriota), and OTU18053 (Proteobacteria) had the highest abundances in the network. Similar seasonal changes of OTU18562 and OTU17592 in abundance were observed, and the other three OTUs showed similar trends

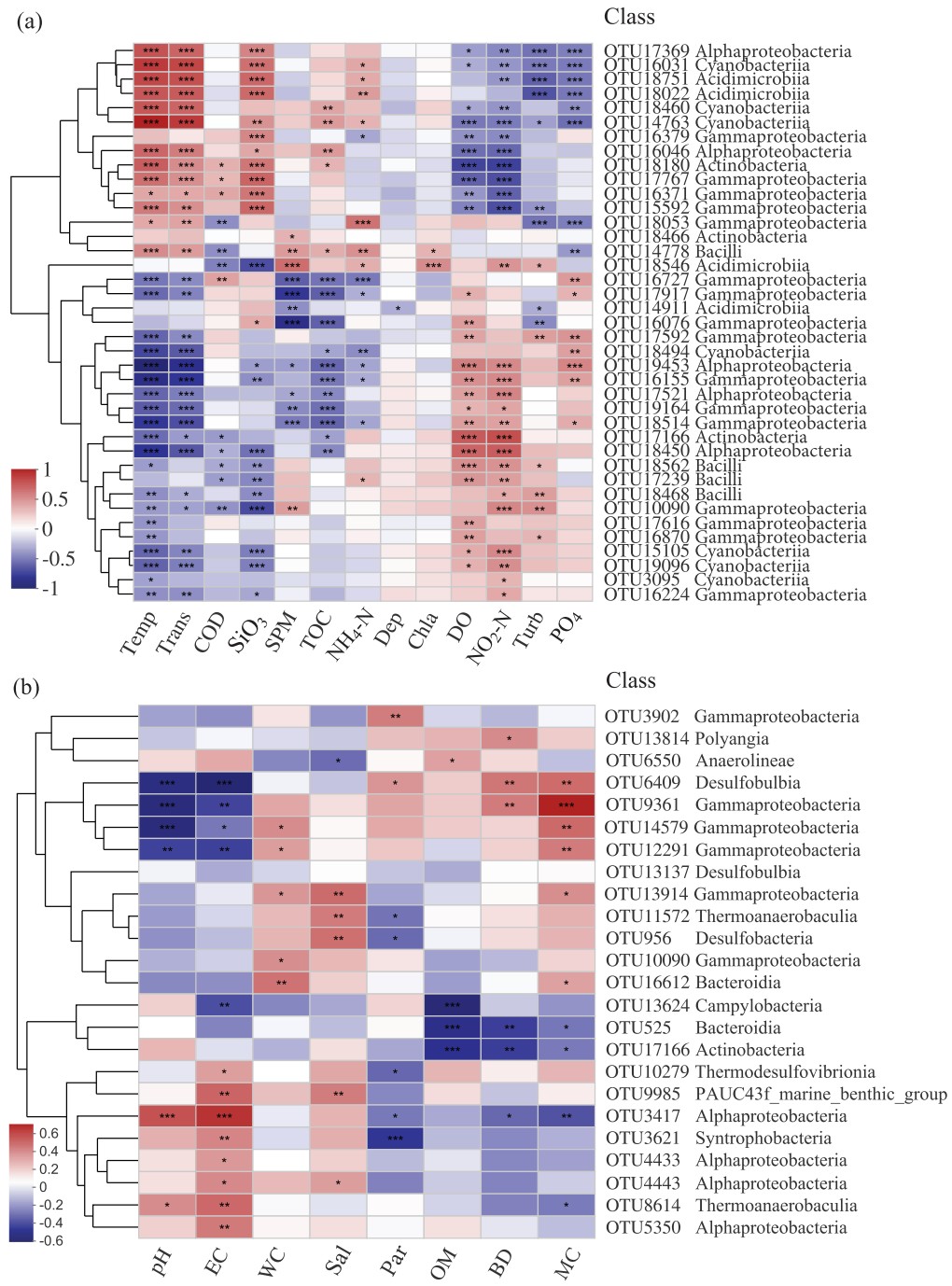

**Figure 5 Heatmap of top abundant OTUs of bacterial communities with environmental factors in the (A) water and (B) sediment of artificial reefs.** The cluster trees were analyzed to show the similarity of OTUs using Bray-Curtis distance. Asterisks represent significant correlations between bacterial OTUs and environmental factors at the following α levels: * = 0.05, ** = 0.01, *** = 0.001. Environmental factors: temperature (Temp); transparency (Trans); chemical oxygen demand (COD); active silicate (SiO₃); suspended particulate materials (SPM); total organic carbon (TOC); ammonium (NH₄-N); depth (Dep); chlorophyll-a (Chla); dissolved oxygen (DO); nitrite (NO₂-N); turbidity (Turb); active phosphate (PO₄); electrical conductivity (EC); water content (WC); salinity (Sal); mean particle sizes (Par); organic matter content (OM); sediment bulk density (BD); mud content (MC).

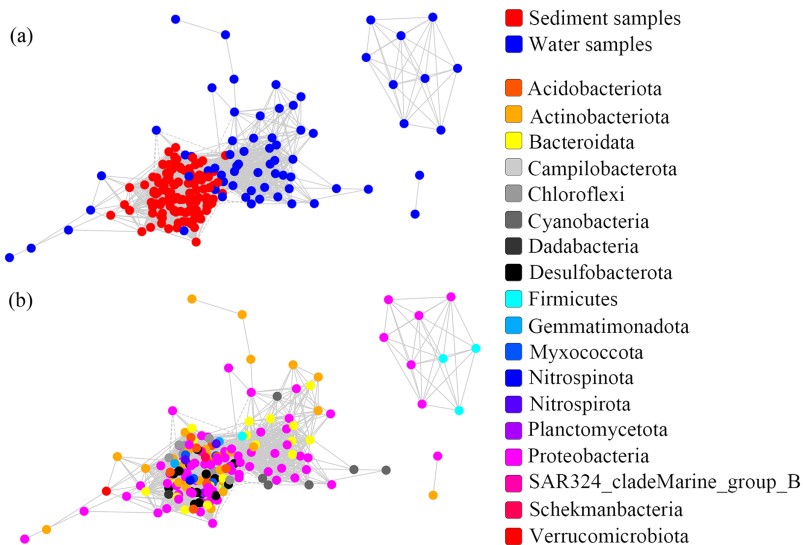

**Figure 6 Co-occurrence networks built from abundant bacterial OTUs in the water and sediment of artificial reefs.** Nodes are colored at (A) habitats and (B) phylum levels. Edges with $|r| \geq 0.8$ and $P \leq 0.001$ are shown in the networks. Positive and negative lines are represented by solid lines and dotted lines, respectively.                                        

(Fig. 7B). OTU18562 showed a most apparent difference among four areas compared with other four abundant OTUs (Fig. 8B). For the sediment bacterial network, 274 nodes belonging to 24 phyla and 1,166 edges were observed (Fig. S5). The top five most abundant OTUs in this network were OTU10090 (Proteobacteria), OTU3621 (Desulfobacterota), OTU13814 (Myxococcota), OTU13137 (Desulfobacterota), and OTU6550 (Chloroflexi). The seasonal variations of the five OTUs were not evident (Fig. 7C), and OTU10090 showed lower abundances in areas RR and CR than in TA and AA (Fig. 8C).

The top keystone OTUs with relative abundances higher than 1‰ were identified as the keystone taxa in the co-occurrence networks. In the total bacterial co-occurrence network, the top 10 keystone OTUs were affiliated to five phyla, Myxococcota, Proteobacteria, Gemmatimonadota, Chloroflexi and Actinobacteriota (Table S2A). The keystone OTU with highest keystoneness was OTU13814 (Myxococcota, keystoneness = 0.840), which ranked 15 among all OTUs with a relative abundance of 12.3‰; the abundance of OTU12321 (Proteobacteria, keystoneness = 0.836) ranked 150 with a relative abundance of 1.8‰. There was no significant seasonal abundance variation for the top five keystone OTUs except OTU13814, and no evident differences among the four areas were found (Figs. S6A, S7A). For the water network, the top 10 keystone OTUs included members of the phyla Bacteroidota, Proteobacteria, Desulfobacterota, Gemmatimonadota and Acidobacteriota (Table S2A). The top two keystone OTUs (OTU525 and OTU7969) had abundances of 2.3‰ (ranked 90) and 1.4‰ (ranked 123), respectively. Similar seasonal trends of the abundance for the top five keystone OTUs were observed, which showed higher abundance in spring and autumn (Fig. S6B). For the sediment network, the top 10 keystone OTUs belonged to three phyla, Proteobacteria, Firmicutes and Chloroflexi (Table S2C). OTU6991 (abundance 2.3‰, ranked 114) and OTU3902 (abundance 8.8‰,

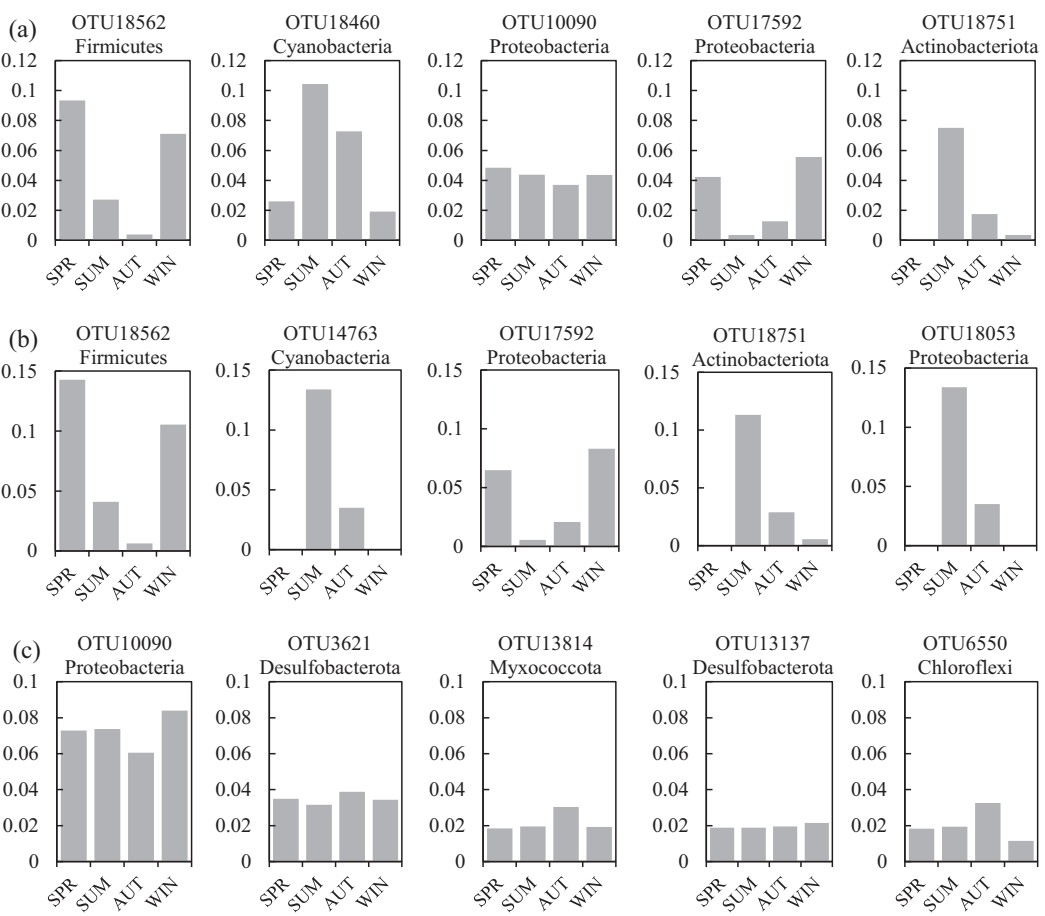

**Figure 7 Seasonal changes of the relative abundance for the top five abundant OTUs of the co-occurrence networks in the (A) water and sediment, (B) water and (C) sediment of artificial reefs.** Four seasons: spring (SPR); summer (SUM); autumn (AUT); winter (WIN).

ranked 24) were the only two keystone OTU with keystoneness greater than 0.7. The changing patterns among the four seasons and four areas for keystone OTU17592, OTU16870, and OTU17616 were consistent (Figs. S6C and S7C).

## DISCUSSION

### Bacterial community characteristics in the water and sediment of ARs

The distributions of the marine bacterial communities were highly of indigenous and specific, while the abundant taxa that represented in the communities were analogous (*Pommier et al., 2007*). In this study, Proteobacteria, Cyanobacteria, and Actinobacteria were the most abundant phyla, with a total relative abundance of 74% in the water samples (Fig. 2), which was consistent with previous studies in coastal waters (*Lee & Eom, 2016*; *Ye et al., 2016*). For the differences of bacterial communities between habitats, 16 shared phyla were observed between the water and sediment, and five phyla among sixteen phyla had higher relative abundances in the water than those in the sediment. The absolute predominance of the phylum Proteobacteria in the water samples may be the main reason.

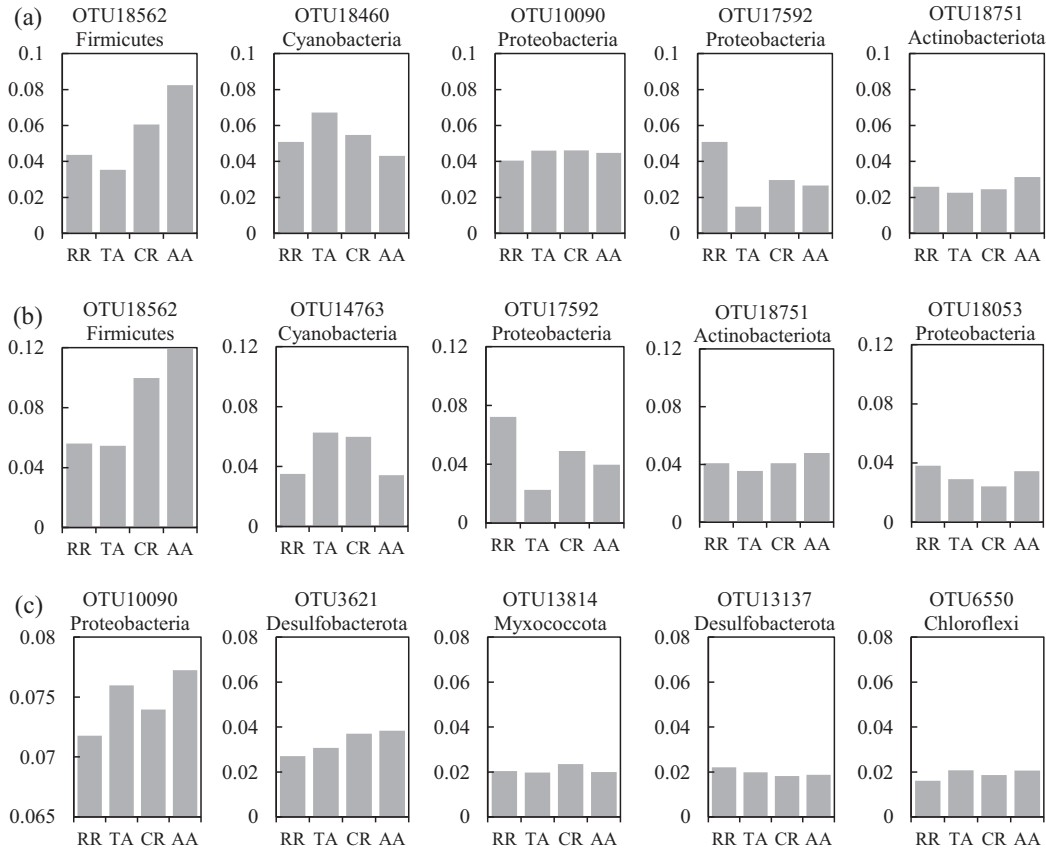

**Figure 8 Variations of the relative abundance for the top five abundant OTUs among four sampling areas of the co-occurrence networks in the (A) water and sediment, (B) water and (C) sediment of artificial reefs (ARs).** Four sampling areas in ARs: rock reefs (RR), transition areas (TA), concrete reefs (CR) and adjacent areas (AA).

Because of the higher abundance of Proteobacteria (42.8%) in the water samples, significantly restricting the population size of the other bacterial phyla (*e.g.*, *Sun et al., 2019*). Higher diversities of the bacterial communities were observed in the sediment than water (Fig. 1; Table 1), which was in accordance with the results of *Feng et al. (2009)* and *Abia et al. (2017)*. Some reasonable assumptions were proposed. *Ye et al. (2009)* stated that sediments within aquatic environments formed more complex environments, which resulted in more prosperous bacterial communities; *Perkins et al. (2014)* noted that sediment provided shelters for bacteria, which helped to defend against the impacts of predation and sunlight; *Liu et al. (2018)* explained that lower concentration of suspended sediment in the water would lead to lower microbial diversity.

The heatmap of the abundant OTUs with four areas revealed that bacterial abundance in RR was inconsistent with other areas, especially in the sediment (Fig. S3). Many studies showed that the diet preference of *A. japonicus* can change the microbial communities in the sediment (*e.g.*, *Zhao et al., 2020*). Thus, we assumed that the aquaculture activities of *A. japonicus* in RR might result in these distinctions. Besides, rock reefs were deployed more intensive than concrete reefs, changing the substrate and flow field surrounding ARs

more significantly (*e.g.*, *Tang et al., 2022*), which might impact the bacterial communities strongly. As for the differences of bacterial abundance among four areas, some inconsistencies were observed. For instance, OTU1283 (Actinobacteriota) showed significantly higher abundance in RR and CR than that in TA and AA, which suggested that it favored ARs. The phylum Actinobacteria plays important roles in mineralizing organic matter in marine sediment (*Bell et al., 1998*), indicating that the organic matter content in the ARs was higher than that in non-intervened areas. OTU7413 (family Rhodobacteraceae) was regarded as an important food resource for *A. japonicus* (*Zhao et al., 2020*), which showed a lower abundance in RR than other three areas.

As for the beta diversity among the four areas, the highest and lowest beta diversities were found in autumn and summer, respectively (Fig. 4). Some current studies highlighted that high beta diversity of bacterial communities in a particular season was a response to environmental heterogeneity (*Fournier et al., 2020*). While some studies addressed that the Bray-Curtis dissimilarities among microbial communities did not significantly correlated with environmental factors such as temperature and salinity (*Balzano, Abs & Leterme, 2015*). These opposite conclusions may result from the differences in spatial and temporal scales (*Hatosy et al., 2013*).

## Seasonal dynamics of bacterial communities in ARs

Marine bacterioplankton exhibit pronounced seasonal succession patterns worldwide (*Bunse & Pinhassi, 2017*), and follow with the changes of temperature, chlorophyll-a, and other nutrients (*Mohapatra et al., 2020*; *Pinhassi et al., 2006*). In this study, seasonal changes in the relative abundances of diversity indices, abundant species and community structures were observed. The seasonal variations of α diversity between water and sediment were inconsistent (Fig. 1). In the water samples, a more considerable fluctuation of α diversity was observed, and had the highest Shannon diversity in spring. Similar seasonal patterns in the Shannon diversity have been observed in other seawaters (*Piwosz et al., 2018*; *Zhu et al., 2020*). Previous studies indicated that sudden decrease of microbial plankton communities in winter and new growing season in spring preceding the spring bloom (*Espinoza-González et al., 2012*; *Figueiras et al., 2020*) could promote these seasonal patterns. Therefore, higher diversity in spring can be explained by the intrinsically high growth rate of bacterioplankton (*Agawin, Duarte & Agustí, 2000*). However, no significant differences of bacterial diversity in the sediment were detected. Marine sediment is characterized as a habitat type with high biodiversity, complicated community structures and spatial heterogeneity (*Brandt & House, 2016*). The bacterial communities in the sediment had wider niche ranges and sufficient nutrients, which helped maintain and facilitate prosperous diversities (*Shu et al., 2020*).

The abundance of the abundant phyla also showed conspicuous varieties (Fig. 2). For the water samples, our results showed that Actinobacteria had the highest abundance in summer, when the salinity was lowest because of higher precipitation level. *Shen, Jürgens & Beier (2018)* confirmed that Actinobacteria is sensitive to the change of salinity.

Meantime, the peak abundance of Cyanobacteria occurred in summer. *Martin, Thomas & Rhena (2017)* also stated that the autotrophic Cyanobacterium, Synechococcus exhibited higher abundances during monsoons. For the sediment samples, the seasonal variations of the abundant bacteria were not significant compared with water samples. For instance, Chloroflexi was the most abundant in autumn. Chloroflexi is regarded as having anaerobic and heterotrophic lifestyles, which are positively related to organic matters (*Wilms et al., 2006*). In autumn, the seaweed cultivation in the adjacent sea could probably explain their higher abundances of Chloroflexi.

Marked seasonality of bacterial community compositions in the water was widely observed. *Shu et al. (2020)* found distinct seasonal patterns of core and non-core bacterial communities in an urban river; *Mohapatra et al. (2020)* discovered that seasonal variations of environmental drivers highly impacted the bacterial communities. However, the seasonal variations of microbial community compositions in the sediment were controversial. We observed significant seasonal changes of bacterial communities at OTU level (Table 2), while some similar studies have presented different conclusions. *Liu et al. (2015)* revealed noticeable seasonal variations of bacteria community because of environmental heterogeneity; *Palit & Das (2020)* reported a comparatively low seasonal fluctuation of sediment bacterial communities in part of sampling sites based on the culture-dependent method; *Ming et al. (2021)* indicated that no marked seasonal difference was observed among the bacterial community compositions. Different temporal and spatial scales may cause these inconsistent results, and more follow-up studies are suggested.

## Environmental characteristics and bacterial communities

Environmental factors significantly affected bacterial communities, such as nitrite, temperature in the water (*Fadeev et al., 2018*; *Zorz et al., 2019*) and mud content in the sediment (*Lee et al., 2020*). In the present study, the Mantel test revealed that nitrite, temperature, and transparency significantly correlated with the communities in the water, while mud content was the only related factor in the sediment (Table 3). Although N and P are limiting factors of eutrophication, they are important nutrients affecting the proliferation of bacteria (*Huang et al., 2017*). Our results revealed that dissolved inorganic nitrogen ($NO_3$-N, $NH_4$-N, $NO_2$-N) was correlated with many abundant OTUs (Fig. 5A). Water temperature is an important factor that governs microbial growth and activity (*Margesin, 2009*), and thirty-three abundant OTUs were significantly related to temperature. Light intensity could also shape different microbial communities (*Davies & Evison, 1991*), and our studies confirmed that the correlation between transparency and the bacterial communities was significant. Different electrical conductivity levels can influence bacteria (*Silverman & Munoz, 1975*), which was in accordance with the results that the abundances of 13 abundant OTUs were affected by EC. Mud content was a vital factor influencing microorganisms from sediment, higher content could offer more nutrition substrates, which was meaningful to support greater microbial biomass (*Wei et al., 2014*).
## Co-occurrence networks and keystone species of bacterial communities in ARs

Although the physical and chemical properties between water and sediment were fundamentally different, the microbial communities are connected through the sedimentation of organic matters (*Lucie et al., 2011*). To reveal the potential interactions among the bacterial communities in two habitats, co-occurrence networks were used to promote our understanding of the microbiomes in marine ecosystems (*Berry & Widder, 2014*). Species similarity between the water and sediment networks was low (Jaccard = 11.8%), and the similarity between the water network and total bacterial network (33.1%) was equal to that between the sediment network and total bacterial network (31.5%). These results indicated that habitats significantly affected the compositions of microorganisms in the co-occurrence networks, which was consistent with *Zhang et al. (2020a)*.

A total of 47 OTUs were simultaneously observed in the three networks, and some played important roles in ecosystems. For instance, family Woeseiaceae (OTU10090) covers a broad physiological spectrum (*Mußmann, Pjevac & Krüger, 2017*), which may promote plant survival by participating in nitrogen and carbon cycling (*Zhang et al., 2020b*); family Methyloligellaceae (OTU3417) was reported as an important host related to antibiotic resistance and metal resistance (*Zhang et al., 2021*). Significant seasonality of the top five abundant OTUs for the networks was exhibited, while the dissimilarities of their abundance among the four areas were relatively low (Figs. 7 and 8). OTU18562 (family Planococcaceae) was the most abundant OTU in the total network and water network, which showed evident temporal and spatial differences, concurrently. OTU18562 affiliates to family Planococcaceae, which was correlated with the decomposition of fertilizers (*Suzuki et al., 2021*). OTU17592 (family Moraxellaceae) had higher abundance in RR and CR than in TA and AA in the total network and water network, indicating positive relationships between OTU17592 and ARs existed. OTU18751 (family Microtrichales), which showed the highest abundance in summer and lowest abundance in spring, played crucial roles in hydrolyzing and utilizing complex organic matters (*Li et al., 2021*).

Keystone species are highly connected taxa in networks that play vital roles in maintaining microbial community structures (*Faust & Raes, 2012*). The top 10 keystone OTUs in the three co-occurrence networks were significantly different. As reported, differences of key microbial groups at ARs and open waters have also been studied (*Zhu et al., 2020*). The ranks of relative abundance for the top 10 keystone OTUs in the three network were low (Table S2). Two factors could explain this phenomenon. On the one hand, it was suggested that abundant bacteria contribute mostly to the biogeochemical cycles, while rare microbiomes might act to stabilize the community (*Shade et al., 2014*; *Genitsaris et al., 2015*). Moreover, rare species may be disturbed by the rapid changes of the environment (*Shade et al., 2014*). On the other hand, rare taxa could considerably increase in abundance to respond the environmental disturbances; and these dynamics may be the reasons for their greater contributions to microbial communities (*Caron & Countway, 2009*).

In general, relative to the important functions of keystone species in the community, the discovery of these species evidently falls behind (*Palit & Das, 2020*), which leads to a limited understanding of microbial ecology. The abundance of the top five keystone OTUs in the water network all showed similar seasonal variations (Fig. S6B), suggesting keystone species were more consistent in response to the changes of environmental conditions than abundant species (Fig. 7B). OTU525 belongs to family Flavobacteriaceae, which were reported as important bacterial populations associated with algal blooms closely (*Pavlovska et al., 2021*). What's more, the seasonality of the top five keystone OTUs in the sediment network was also significant (Fig. S6C). Three keystone OTUs (OTU17592, OTU16870 and OTU17616) in the sediment network had significantly higher abundance in RR than in other three areas (Fig. S7C), showing their positive relationships with the deployment of rock reefs. These three keystone OTUs all affiliate to order Pseudomonadales, class Gammaproteobacteria. As reported by *Eswaran & Khandeparker (2019)*, Pseudomonadales participated in mediating the degradation of carbohydrates by producing β-Glucosidases in a tropical estuarine environment. Some other keystone OTUs are also crucial in the particular ecological process. For instance, the abundance of OTU7413 (family Rhodobacteraceae) ranked 137th in the water network (Table S2B), which was regarded as the key member of the initial microbial biofilm in coastal seawater (*Hila et al., 2013*). OTU11572 and OTU8614 belong to family Thermoanaerobaculaceae, may play crucial roles in nitrogen transformation in water (*Wang et al., 2019a*).

## CONCLUSIONS

In the present study, we simultaneously profiled the temporal and spatial variabilities and potential interaction patterns of bacterial communities in the water and sediment of ARs for the first time. Seasonal variations of bacterial community compositions were observed in the two habitats, while spatial changes were only detected in the sediment. Bacterial communities in the rock reef area were significantly different with other three areas, which indicating the deployment of ARs impacted the bacterial communities in sediment. The potential correlations among bacterial communities in the sediment were lowest in the three networks, suggesting less niche overlap existed. The abundant and keystone species in the networks showed discordant variations among four seasons and four areas, revealing that two types of species played different ecological functions.

As important artificial habitats in marine fisheries, our discoveries about ARs are crucial to better comprehend the mechanisms of microbial ecology in artificial habitats, to promote restoration efficiency, to improve biodiversity, and to recover the marine environment. Particularly, marine microbial loops are comprehensive networks containing bacteria, archaea, and eukaryotes, and further studies are suggested to reveal the interactions among these microbiomes in artificial habitats.

## ACKNOWLEDGEMENTS

We thank all scientific staff and crew members of Qingdao Longpan Company for their assistance in the surveys.

### Funding

This study was funded by the Project of Investigation of Fishery Resource of Marine Ranching from the Department of Agriculture and Countryside of Shandong Province. The funders had no role in study design, data collection and analysis, decision to publish, or preparation of the manuscript.

### Grant Disclosures

The following grant information was disclosed by the authors:
Project of Investigation of Fishery Resource of Marine Ranching from the Department of Agriculture and Countryside of Shandong Province.

### Competing Interests

The authors declare that they have no competing interests.

### Author Contributions

- Guangjie Fang conceived and designed the experiments, performed the experiments, analyzed the data, prepared figures and/or tables, authored or reviewed drafts of the paper, and approved the final draft.
- Haolin Yu performed the experiments, analyzed the data, authored or reviewed drafts of the paper, and approved the final draft.
- Huaxiang Sheng performed the experiments, prepared figures and/or tables, and approved the final draft.
- Chuanxi Chen performed the experiments, authored or reviewed drafts of the paper, and approved the final draft.
- Yanli Tang conceived and designed the experiments, authored or reviewed drafts of the paper, and approved the final draft.
- Zhenlin Liang analyzed the data, authored or reviewed drafts of the paper, and approved the final draft.

### Field Study Permissions

The following information was supplied relating to field study approvals (*i.e.*, approving body and any reference numbers):
Field experiments were approved by the Ministry of agriculture and rural areas of China.

### Data Availability

The sequencing data are available in the National Center for Biotechnology Information (NCBI) Sequence Read Archive: PRJNA725051.
The environment factors are available in Table S1.

## Supplemental Information

Supplemental information for this article can be found online at http://dx.doi.org/10.7717/peerj.12705#supplemental-information.

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
