# Peer review of "Seasonal variations and co-occurrence networks of bacterial communities in the water and sediment of artificial habitat in Laoshan Bay, China"

_PeerJ, doi:10.7717/peerj.12705_

## Round 0.1 · original submission · Major Revisions

Both reviewers agree that the manuscript presents interesting results that should be published in PeerJ after some changes have been done. I have decided for "major revisions" mainly because the reviewers have asked for some significant modifications in the figures, but they could otherwise be considered minor revisions.

Please read carefully the reports by the two reviewers, including the additional files, and try to address their suggestions and concerns.
As mentioned by the reviewers, the criteria for what you are considering "dominant species" should be clearly defined.

Moreover, please follow the guidelines of the Bacteriological Code for nomenclature: genera and species names must be written in italics.
I am looking forward to reading a new version of your manuscript.

Reviewer 1 ·

Basic reporting

The article in general is understandable but some changes need to be done for a better understanding.

The article is very well supported with references. However, the way in which some of these references are cited needs to be checked, and some references are missing.
In particular, in the discussion section I recommend the authors to try to use better the references in the context of their study. In some cases, they just mention the results of previous studies. I encourage them to try to link all this information.

In general, the structure of the article follows the standards of the Journal and the study sequences have been deposited in NCBI for public accessibility.
The results presented are mostly supported by figures or tables, with some exception. New figures need to be performed to support some results mentioned in the manuscript, while some others should be re-done and displayed in a clearer way.

Experimental design

The authors take the most of the sequencing technologies to bring a step forward the study on artificial reefs from a microbiological perspective.

In principle, the experiment is well designed, but more explanation on the co-occurrence networks analyses and clarifications are needed. For example in the study design, the number of samples, replicates, and samples used in networks analyes need to be cleared in the manuscript. Moreover, I recommend the authors to explain the theresholds used to define dominant bacteria, and the reasons why they decided to based their research only on this fraction of the bacterial community.

Validity of the findings

The analyses are appropriate for this research. However, some missing information is needed and the interpretation and discussion of these results is confusing. Moreover this issue extends also to the conclusion, which in principle is well stated and connected to the original question of the research, but it is based on these confusing results,thus losing its validity.

Annotated reviews are not available for download in order to protect the identity of reviewers who chose to remain anonymous.

Reviewer 2 ·

Basic reporting

Even though I’m not English native speaker, I think that the MS is clear on the use of English.
In the abstract section the most relevant data found should be showed. Bacterial communities must be displayed among sampling sites (habitats) to understand possible differences.
Introduction and background sections are well presented to show their finding within the proposed study field, with adequate and relevant literature.
Materials and methods showed clear definition of sampling site, design structure and data analysis. Some clarifications are needed, which are detailed in the correspond section of this review.
In results, relevant data are showed. However, does not follow a clear pattern from previous section. Some adjustments may be necessary to provide a better progress of results from applied methodologies.
Regarding to conclusions, authors should stress out the importance of dominant bacterial communities, specially associated with their ecological role in both water and sediment samples.
Figures and tables have well definition and are relevant for the aim of this MS. Its necessary include some elements in legends.
Support data were revised and easily find out.
In general, the MS provide valuable information, with plausible relationship among habitats for bacterial communities and environmental factors. Especially for the last ones, which it is hard to find out.

Experimental design

This section was well performed, just need some clarifications to be completely understanding.

Validity of the findings

All findings are relevant, especially because contemplate interesting variables such as environmental factors, examination of bacterial communities from microbiological data for different habitats.
Conclusions should be better oriented to understand why some bacterial communities are present in evaluated areas. Probably a good explanation of their ecological role could express better their findings.

Additional comments

General comment of the MS
This MS provide valuable information, with plausible relationship among habitats for bacterial communities and environmental factors. Especially for the last ones, which it is hard to find out.
All editorial criteria were revised.
The current MS should be improved upon before acceptance. Some comments and suggestions are below detailed.


Abstract
In this section more robust data are need, please show valuable data that you have reach to give robustness to this section. Also, bacterial communities must be displayed among sampling sites (habitats) to understand possible differences.

Introduction
This section is well presented, with adequate literature.
Specific point:
Line 51 (Sun et al., 2017), please check publication year. Its does not match with reference (2019)

Material and methods
Here its possible to find a clear definition of sampling site, design structure and data analysis.
However, some clarifications are need.
Specific points
Lines 104-106. First, within samples for water and sediment among season are 24, according to description provided. Then, the number of 32 samples are not clear from where it comes.
Line 111. May its necessary to include 1L instead of other part.
Line 113. Is necessary to describe why a grab sediment sampler was used. This tool is commonly used but generates some resuspension of sediment particles, that might could affected bacterial taxa identification in sediment samples.
Line 125. Check publication year of (State Bureau of Quality and Technical Supervision of China, 2007), its does not match with reference (2001).

Results
This section showed relevant data. However, does not follow a clear pattern from previous section. Some adjustments may be necessary to provide a better progress of results from applied methodologies.
Specific points
Line 190. Table 1 is mentioned, but from this table its not possible determine differences among habitats.
Line 193. May the word “much” should be removed
Line 194. Its possible to stress that a seasonal pattern could be observed.
Line 197-198. Seasonal variations of phylum are not clear as described (“evident”) in these lines, especially for those collected from sediment samples. Also, areas are used as check point. I think that are you talking about seasons?
Line 203-207. The fact that 4 phylum on water samples counts for over 70% of total abundance should be incorporated. The same facts should be stressed for sediment samples. Then this could be a first approach to diversity among sites
Line 206. Given percentages are associated to total abundance?
Line 207. Why Cloroflexi is particularly mentioned?
Line 210. Table 2 does not show mentioned facts.
Line 214. Include bacterial within “ten classes”.
Line 215-217. An explanation of how or why described differences of bacterial communities among areas were found.
Line 227-228. More than OTUs explanations, classes must be stressed out. May be is better to explain that 3 mentioned OTUs belongs to a clade in sediment, for example.
Line 237-240. Described OTUs abundance among seasons and networks are not easy to understand because there is no data available to figure out.

Discussion
In this section a good discussion could be find out. However, I think that bacterial communities and their interaction with habitats must be highly described, to understand why some communities are present or not in evaluated areas.
Specific points
Line 270. “Highly of endemism” should be well described.
Line 276. What “inconformity” means?
Line 276-278. Firmicutes abundance is well described among areas, but its ecological role does not, then just a simple result is showed instead of a discussion (in concordance with the general comments of this section).
Line 287-289. Re-write this paragraph, because it’s difficult to understand why 5 phyla in water had high abundance compared with phyla from sediment samples. I can´t see the relation with 16 shared Phylum. Then, why the abundance of Proteobacteria is described as a main factor to explain the above-mentioned facts. Again, the ecological role of this phylum should provide a better idea about it.
Line 292. “Some assumptions were reasonable”. I don’t understand this sentence.
(Ye et al., 2007), publication year must be verified from reference, does not match.
Line 304. (Margesin et al., 2009) is not in reference. (Margesin, 2009)?
Line 336. What do you want to say with “sufficient nutrition”?
Line 338-339. (Brandt et al., 2016) is not in reference. (Brandt and House, 2016)?
Line 371. What do you want to say with “mature”? could you explain this term. Do you want to say something like stable?
Line 376. (Faust and Raes et al., 2012), delete “et al.”
Line 386. (Caron et al., 2009) is not in reference. (Caron and Countway, 2009)?

Conclusions
Here authors should stress out the importance of dominant bacterial communities, regarding to it ecological role in both water and sediment samples.
Line 403. “mature” term must be clarified.

Reference
Some ones must be revised
Lines: 443, 448, 469, 471, 530, 603, 606.

Figures
Fig 1. To provide a better comparison point, Shannon index values from sediment should be located over the ones from water. Also values and bars (mean and SD?) should be explained in the legend of this figure.
Fig 3. Principal coordinates analysis (PCoA) plots of bacterial communities in the (a) water and (b) sediment of artificial reefs. Why fig 3a has water and sediment samples?
Fig 6. In the legend, “Positive and negative lines are represented by the gray solid lines and black dotted lines.”. Respectively?, should be included after “dotted lines, ”

Tables
Some extra information should be included in legends, same as in supplementary figures and tables.

Table 1. P values, what it means?. It corresponds to differences among seasons for each index.
Table 2. Describe what is F, R and P values.
Table 3. Describe what is r and P values.

Fig S1. Describe what are CR, RR, TA and AA

---

## Round 0.2 · Minor Revisions

Both reviewers and I agree that the manuscript has been significantly improved and it is almost ready to be acceptable for publication. They are suggesting a number of minor changes and there are some minor issues with the terminology. Please address these changes and submit a new version. Please notice that the suggestions by Reviewer #2 are in the attached annotated manuscript.
I will gladly agree to accept the corrected version for publication as soon as possible after I received the minor corrections.

Reviewer 1 ·

Basic reporting

The authors have incorporated many changes making the manuscript more understandable. However, there are still a few details they should pay attention on.
For example, they still use the term "interaction" in the abstract and the Keywords sections, and I would recommend the authors to change it by correlation or at least specify that they mean potential interactions.

Experimental design

I appreciate the authors did clarify many things they were asked to.
Nevertheless, there is some inconsistencies related to the networks analyses. First of all (lines 238 and next), they said that the networks were built in Cytoscape, but they don't mention the tool in Cytoscape for doing so. Maybe they want to say that they used Cytoscape to visualize the networks.
Next, they explain clearly how they calculate the Spearman correlation and how they selected the significant correlations, and they cited a paper by Cheung et al. (2018). However, in this paper, Cheung et al. calculated the networks using the SparCC R package. Either way, calculating "manually" the correlation networks or by using the SparCC R package is correct. But I recommend the authors to be clear in this part, and whatever the way they calculate the networks was they have to explain it and cite it properly.

Moreover, in the Results, in line 337 the authors introduced the concept of "the most abundant OTUs in the network". It is not clear what do they mean with this. Each of the OTUs usually appear once in the ecological networks representing one single node. I believe that the authors mean the OTUs in the networks that presented the highest abundance. In any case, they should either rephrase or explain how they defined the most abundant OTUs in the network.

Finally, in line 357 they wrote "the capital keystone". I recommend the authors to define what is it the capital keystone.

Validity of the findings

no comment

Additional comments

In line 61 replace "which" by "being"
Sentence in lines 77-78 is not clear
Line 185 change "was" by "were"
Line 160 Spearman with capital S
Line 354 replace "of more" by "higher"
Line 358 remove "which"
Line 412: what do they mean by "five-sixteenths phyla"?
Line 415: replace "which" by "might"
Line 535: I think the authors mean "needed" instead of "suggested"
Line 546: change "varied" by "variation in"
Line 551: the authors said "influencing microorganisms". I recommend they specify if they influenced microorganisms from sediments or from seawater.
Line 586: Indicate the lowest taxonomical rank the OTU17592 belongs to, as done for the others.
Line 619: at the end of the sentence that stated that "rare microbiomes might act to stabilize the community", please indicate the reference.
Line 658: ARs "are" an important artificial habitats in marine "fisheries"
Moreover, the reference they include to cite Cytoscape is not correct, and they should include the one provided by https://cytoscape.org/

Reviewer 2 ·

Basic reporting

Suggestions were carefully considered, especially in abstract, results and conclusions.
Also, modifications of figures and tables has been considered, complemented with some new or modified figures, that reach better understanding of this MS.
In general, the MS provide relevant information within ARs habitats for bacterial communities and environmental factors.

Experimental design

This section was well performed since first draft, with clear and accurate clarifications of referee's observations

Validity of the findings

Results are relevant, especially because contemplate interesting variables such as environmental factors, examination of bacterial communities from microbiological data for different habitats among ARs.
Adjustments on results and discussion sections has been highly improved

Additional comments

The current MS version correctly reach some previous observations and provide to readers a better description of interesting results.

Annotated reviews are not available for download in order to protect the identity of reviewers who chose to remain anonymous.

---

## Round 0.3 · accepted · Accept

All minor issues have been satisfactorily addressed. Congratulations.